# A More General Quantum Credit Risk Analysis Framework

**DOI:** 10.3390/e25040593

**Published:** 2023-03-31

**Authors:** Emanuele Dri, Antonello Aita, Edoardo Giusto, Davide Ricossa, Davide Corbelletto, Bartolomeo Montrucchio, Roberto Ugoccioni

**Affiliations:** 1Dipartimento di Automatica e Informatica (DAUIN), Politecnico di Torino, 10129 Torino, Italy; 2IBM Italia, 20090 Milano, Italy; 3Intesa Sanpaolo, 10121 Torino, Italy

**Keywords:** quantum computing, algorithms, scalability, credit risk analysis, quantum finance

## Abstract

Credit risk analysis (CRA) quantum algorithms aim at providing a quadratic speedup over classical analogous methods. Despite this, experts in the business domain have identified significant limitations in the existing approaches. Thus, we proposed a new variant of the CRA quantum algorithm to address these limitations. In particular, we improved the risk model for each asset in a portfolio by enabling it to consider multiple systemic risk factors, resulting in a more realistic and complex model for each asset’s default probability. Additionally, we increased the flexibility of the loss-given-default input by removing the constraint of using only integer values, enabling the use of real data from the financial sector to establish fair benchmarking protocols. Furthermore, all proposed enhancements were tested both through classical simulation of quantum hardware and, for this new version of our work, also using QPUs from IBM Quantum Experience in order to provide a baseline for future research. Our proposed variant of the CRA quantum algorithm addresses the significant limitations of the current approach and highlights an increased cost in terms of circuit depth and width. In addition, it provides a path to a substantially more realistic software solution. Indeed, as quantum technology progresses, the proposed improvements will enable meaningful scales and useful results for the financial sector.

## 1. Introduction

This paper is intended as an extension of previous work [1]. In particular, this extended version includes new data, analysis, and theoretical developments not present in the original paper and that derive from recently acquired access to 7- and 27-qubit QPUs. The authors believe that these additions provide significant new insights and perspectives regarding the enhanced algorithm and its practical implementation.

### Quantum Finance and Credit Risk Analysis

The field of quantum finance aims to use quantum computing to solve a variety of computational problems in finance more effectively than classical methods [2,3]. In recent years, researchers have focused on achieving a quantum advantage in credit risk analysis (CRA) [1,4]. CRA is a crucial risk management tool that assesses the risk of loss from a debtor’s insolvency [5]. Classically, Monte Carlo methods are commonly used in the field to estimate economic capital, which is the amount of capital needed to ensure a company remains solvent based on its risk profile. Essentially, these estimation techniques depend on obtaining numerical results through repetitive random sampling [6]. A practical example of their utilization is the computation of the value at risk (VaR), a statistic that quantifies how much a set of investments might lose (with a given probability) over a defined time-frame [7]. This metric is broadly used for the assessment of EC, but in most cases, no closed-form solution currently exists for computing it [8].

However, Monte Carlo simulations are computationally expensive due to the rare-event simulation problems inherent in credit risk evaluation [9]. Additionally, Monte Carlo simulations can only generate pseudo-random variables, and the quality of the simulation can be compromised by the appearance of patterns [10].

To overcome these limitations, researchers have explored new methods, such as those based on quantum computing, which can naturally generate true random samples due to the probabilistic nature of qubits [11]. Moreover, quantum amplitude estimation (QAE) has shown promise in estimating the value at risk and offers a quadratic speedup over classical Monte Carlo methods [4,12].

However, the existing quantum algorithm for CRA [4] is based on the Basel II framework, built on an ASFR (asymptotic single-factor risk) model [13], which assumes a borrower will default if the value of its assets falls below the value of its liabilities [13]. A visual representation is provided in Figure 1. While this model is useful, it is not optimal, especially for complex credit risk portfolios. In fact, though it helps to reserve an EC amount that suits every default scenario, it is intended to be a standard tool for CRA and therefore it is deliberately conservative [14]. Large financial institutions use custom models that consider several risk factors instead of just one since this refinement allows them to reserve a more precise amount to cover potential losses [15]. To address this aspect, we proposed modifications to the existing quantum algorithm to handle increased complexity in the assets’ default model, while preserving the advantages of quantum computing in terms of needed (quantum) samples.

Additionally, we presented a solution to encode non-integer values for the loss-given-default input parameter to use real-world data and provide a fair comparison with traditional benchmarks.

We are now able to provide experimental results for the enhanced version of the original quantum algorithm, not only through classical simulation of quantum hardware but also from cloud access to IBM QPUs with 7 and 27 superconducting qubits.

In the following sections, we first introduce the use of quantum amplitude estimation for CRA. We then present the proposed modifications to the existing algorithm to address outstanding issues, including the Basel II model’s limitations and the encoding of non-integer values. Lastly, we present the new results of simulation experiments and of executions on real quantum devices obtained by running the experiments on IBM devices from the *Researchers* program [16] and from the *pay-as-you-go* service [17].

## 2. Methods

As described in [1], credit risk can be evaluated through three primary measures: the probability of default (PD), the loss given default (LGD), and the economic capital ($E_{cap}$). The PD represents the likelihood that the debtor will become insolvent, while the LGD is the estimated loss following the insolvency of the counterparty. The expected loss is another commonly used risk measure, which depends on both the PD and LGD, as an increase in either quantity results in a higher expected loss. Multiplying the PD and LGD values gives the expected loss for each exposure. This measure is additive, so the expected loss for a portfolio of *n* assets is the sum of each exposure’s expected loss.
(1)$$E\left[L\right]=\sum_{k=1}^{n}PD_{k}\cdot LGD_{k}$$

$E_{cap}$ is the third measure used to assess credit risk. It is defined as the amount of equity that a financial institution will maintain to manage the risk of credit losses in its portfolio. The economic capital, which is the VaR (quantile of losses at a certain confidence level α) minus the total expected loss, is determined based on the distribution of losses.
(2)$$E_{cap}=VaR_{\alpha }-E\left[L\right]$$

The expected loss is already taken into account in financial reports for financial institutions, so it is subtracted from the VaR and thus not factored into the EC. Therefore, the economic capital is used to measure unexpected or extreme values of losses rather than average losses.

### 2.1. SOTA Quantum Credit Risk Analysis

The quantum amplitude estimation (QAE) algorithm [18] provides a quadratic speedup compared to classical Monte Carlo methods [12]. QAE has been utilized to determine VaR in prior research [4]. A variant of QAE called iterative QAE (IQAE) has recently been proposed as well [19]. This variant reduces the number of required qubits and gates while maintaining the quadratic speedup (up to a logarithmic factor) over classical methods.

In order to exploit the speedup guaranteed by the QAE algorithm, the problem under consideration has to be mapped to a Hermitian operator A acting on n+1 qubits. This A operator is constructed in the following way:(3)$${A|0\rangle }_{n+1}=\sqrt{1-a}{|\psi_{0}\rangle }_{n}|0\rangle +\sqrt{a}{|\psi_{1}\rangle }_{n}|1\rangle$$
where a∈[0,1] represents the probability of measuring the last qubit in the quantum state |1〉. The last qubit is in fact the one identifying the property of interest. The QAE algorithm permits us to effectively estimate the value of *a*. The reader can refer to [4,12,18] for additional information on QAE.

In prior research [4], QAE has been utilized to determine the cumulative distribution function (CDF) of the total loss L and construct a Hermitian operator A such that a=P[L≤x] for a given x≥0. Then, a bisection search is applied in order to locate the smallest $x_{\alpha }\geq 0$ such that $P[L\leq x_{\alpha }]\geq \alpha$, implying that $x_{\alpha }=VaR_{\alpha }$. Thus, the aim when calculating $VaR_{\alpha }$ is to identify the minimum threshold for which the estimated probability is greater than or equal to α.

To map the CDF of the total loss to a Hermitian operator A, three operators are usually required:U, which loads the domain-dependent uncertainty model.S, which computes the total loss over $n_{S}$ qubits.C, which flips a target qubit if the total loss is equal to or lower than a certain threshold *x*.

Operator C is used to execute the bisection search needed to compute the VaR.

For what concerns the default model, the framework implemented in [4] is similar to the Basel II internal-ratings-based (IRB) method known as the Gaussian conditional independence model [20,21]. In compliance with this model, all losses can be represented by $L_{k}=LGD_{k}\cdot X_{k}$, where $X_{k}\in \left\{0,1\right\}$ is a related Bernoulli random variable. The probability for asset *k* to default is the probability that $X_{k}=1$. According to the Basel II approach, assuming a latent random variable Z (also referred to as a systemic risk factor) with a realization *z*, the Bernoulli random variables $X_{k}|Z=z$ are considered independent. However, their default probabilities $PD_{k}$ depend on *z* while Z adheres to a standard normal distribution. The default probability $PD_{k}\left(z\right)$ is given by
(4)$$PD_{k}\left(z\right)=F\left(\frac{F^{-1}\left(p_{k}^{0}\right)-\sqrt{\rho_{k}}z}{\sqrt{1-\rho_{k}}}\right)$$
where $p_{k}^{0}$ represents the default probability for z=0, *F* represents the CDF of the standard normal distribution, and $\rho_{k}\in \left[0,1\right)$ determines the sensitivity of $X_{k}$ to Z [4].

### 2.2. Multiple Risk Factors

In the original single-factor model presented in [4], the default probabilities of the counterparts are encoded in a qubit register on which one Y-rotation $R_{Y}\left(\theta_{p_{0}}^{k}\right)$ per qubit is applied with angle $\theta_{p_{0}}^{k}=2\arcsin\left(\sqrt{p_{k}^{0}}\right)$. These rotations comprise the loading operator U introduced in [4].

The original implementation also makes use of a register with $n_{Z}$ qubits. This register encodes a truncated and discretized version of Z using the method proposed in [22]. In this way, we include systemic risk in the quantum uncertainty model, using the realization of Z to prepare the qubits representing the counterparties through controlled rotations with angles $\theta_{p}^{k}\left(z\right)=2\arcsin\left(\sqrt{PD_{k}\left(z\right)}\right)$.

As stated in Section 1, the single-factor model, as implemented in the Basel II framework, is intentionally designed to be conservative [14]. However, it has been recognized that this model has limitations, prompting large financial institutions to seek alternatives to measure risk more accurately. The most common approach extends the single-factor model and employs multiple systemic risk factors. This extension aims to directly attribute default correlations and probabilities to the risk factors, thereby capturing a more realistic depiction of credit risk. The extended model provides a significant advantage by utilizing real-time information about the credit cycle, which enhances the accuracy of the underlying credit risk assessments. As a result, this approach represents a fundamental departure from the uncorrelated defaults inherent in the base model, and it can capture the linkages between economic and financial market factors. In light of these considerations, it is evident that the extended model constitutes a valuable tool for improving risk management practices in the financial sector, particularly for assessing the credit risk of large and complex financial institutions. Furthermore, this approach can reduce uncertainties about the parameters needed for portfolio models’ value-at-risk calculations [15], which is particularly critical for risk-sensitive regulatory capital requirements. Thus, this extended model is widely adopted as a tool for more accurate risk measurement in the financial sector.

In the proposed implementation of the model described above, each risk factor $Z_{i}$ still adheres to a standard normal distribution and presents a weight $\alpha_{i}$ computed by financial institutions, taking into account possible correlation effects among the different factors considered [23]. Therefore, the default probability depends on a random variable Y, which is a linear combination of the *R* risk factors considered.
(5)$$Y=\sum_{i=1}^{R}\alpha_{i}Z_{i}$$

From a practical standpoint, this model comprises multiple latent random variables whose realizations, when appropriately combined, determine the probability of default for each asset.
(6)$$PD_{k}\left(z\right)=F\left(\frac{F^{-1}\left(p_{k}^{0}\right)-\sqrt{\rho_{k}}\sum_{i=1}^{R}\alpha_{i}z_{i}}{\sqrt{1-\rho_{k}}}\right)$$

However, with increased complexity comes the need for alternative approaches to implement a quantum multi-factor version of the canonical uncertainty model. To address this challenge, we proposed two alternatives, each with unique advantages and limitations.

The *first alternative* for encoding systemic risk factors in quantum financial applications involves the use of multiple quantum registers. In this approach, each systemic risk factor, denoted as $Z_{i}$, is assigned its own register, with the values in these registers corresponding to multiple normal standard distributions. The realization of these distributions controls one linear rotation for each asset in the portfolio, with each rotation being weighted by the corresponding $\alpha_{i}$ using the slope of the rotation. This produces a set of rotations that are used to encode the default probability of each asset in the portfolio, as in the original algorithm. The circuit corresponding to this process is illustrated in Figure 2.

While this alternative only requires a limited number of extra qubits to represent the various risk factors, it does entail a significant increase in the number of gates required to implement the encoding process. Specifically, each additional risk factor considered will necessitate *K* new controlled linear rotations. This increased number of gates is because each risk factor requires a separate register and rotation, which in turn requires additional controlled operations in the circuit.

The *second alternative* for implementing a quantum multi-factor version of the canonical uncertainty model involves a single quantum register encoding a random variable N that follows a multivariate normal distribution. A sum register is employed to add up the values taken by the normal distributions, corresponding to the marginal distributions of the multivariate distribution, with each marginal distribution representing a risk factor. The resulting value is used to perform a single linear rotation for each asset, to encode its default probability in the target qubit. The circuit corresponding to this second process is illustrated in Figure 3.

However, since a single rotation is performed per asset, accounting for all the risk factors, the multivariate normal distribution, in this case, is non-standard. This is because it is not possible to encode the weights in the slope of the rotations. Instead, the covariance matrix of the distribution is used to encode the α weights. This approach has a significant drawback, as it requires the same α vector for all the assets.

Despite this limitation, this approach reduces the circuit depth compared to the previous one, as only one rotation is required for encoding the asset’s default probability. However, the method incurs overhead in terms of the required qubits due to the presence of an extra sum register. Nevertheless, this overhead becomes negligible in a scenario with portfolios composed of thousands of assets.

For a more detailed evaluation of the qubits and gates required by the various approaches, we refer the reader to Section 4.1.

Both multi-factor approaches provide an advantage over the Basel II single-factor model by using actual information about the point in time of the credit cycle. Uncertainties about the parameters needed for value-at-risk calculations in portfolio models can thus be reduced.

### 2.3. Arbitrary LGD

One limitation of current implementations of quantum credit risk algorithms is the constraint on LGD parameters, which can only assume integer values due to the use of a weighted sum register in the operator S that computes the total loss. The function S operates as follows:(7)$$\begin{array}{ccc} \; & S:{|x_{1},\ldots ,x_{K}\rangle }_{K}{|0\rangle }_{n_{S}}\;\; & \mapsto {|x_{1},\ldots ,x_{K}\rangle }_{K}{|LGD_{1}x_{1}+\cdots +LGD_{K}x_{K}\rangle }_{n_{S}} \end{array}$$

Here, $x_{k}\in \left\{0,1\right\}$ denotes the possible realizations of $X_{k}$, while the loss given default of each asset is implemented using the weights of the *WeightedAdder* register provided by Qiskit [24,25], which are limited to integer values. We also require $n_{S}=\left\lfloor \log_{2}\left(LGD_{1}+\cdots +LGD_{K}\right)\right\rfloor$+1 qubits to represent all possible values of the sum of losses given default in the second register.

This constraint is particularly limiting considering the small number of currently available qubits. For instance, using three assets with LGD values in the order of $10^{5}$, around 20 qubits would be needed just for the sum register. To allow for more realistic input data, we proposed an alternative version of the algorithm that eliminates the S operator. In particular, we modified the C operator using a circuit that implements a piecewise linear function $\hat{f}:0,\ldots ,2^{n}-1\to \left[0,1\right]$ on qubit amplitudes [4,26,27]. The modified C operator is defined as:(8)$$F|x\rangle |0\rangle =\sqrt{1-\hat{f}\left(x\right)}|x\rangle |0\rangle +\sqrt{\hat{f}\left(x\right)}|x\rangle |1\rangle$$
where |x〉 is an *n*-qubit state. This new approach allows the operator to directly read defaulted qubits from the X-register and associate them with the corresponding total loss. The objective qubit is flipped only if the total loss is less than or equal to the given level *x* set by the current bisection search step. Essentially, the operator reads the X-register as a binary number, and then the specific total loss associated with that binary number is compared with *x* to determine if the objective qubit should be flipped.

In the next section, we apply this improved algorithm to an illustrative example using both classical simulations of quantum hardware and real quantum computers.

## 3. Results

In this section, we present the results of experiments conducted on toy models that illustrate the proposed improvements.

The chosen numeric values for the LGD parameters demonstrate the increased flexibility allowed by our approach compared to the previous one. Each latent random variable $Z_{k}$ was modeled using two qubits. No qubits were needed for the sum register as it is not required for the proposed algorithm.

### 3.1. Noiseless Simulation

The noiseless experiment utilized the multiple-rotations scheme with K=2 assets and two systemic risk factors (R=2). Table 1 provides the values of the parameters used in the experiments. To simulate the experiment, the circuit for A was supplied to the iterative amplitude estimation sub-routine implemented in Qiskit [24]. We performed the bisection search using the result to find $VaR_{\alpha }$, with α=0.95. For the iterative quantum amplitude estimation, we set a target precision of ϵ=0.002 and a 99% target confidence interval. This resulted in an average of approximately 50,000 quantum samples used by the IQAE algorithm to achieve the desired precision and confidence. The entire experiment required 9 qubits that were first simulated (without noise simulation) on classical computers using the simulation back-ends provided by Qiskit [24]. The resulting loss distribution is displayed in Figure 4. Additionally, Figure 5 shows the corresponding CDF and the target level for the value at risk.

### 3.2. Real Hardware and Noisy Simulations

The experiments described in this section aimed to approach as closely as possible the practical implementation of the algorithm on actual quantum hardware. The results can provide a reference point for future research works that may want to evaluate technological improvements. For this reason, the data used were specifically generated by domain experts who took into account what realistic and reliable values for input measurements could be. All the relevant data are available in a public repository [28].

For these experiments, we tested various configurations of the multiple-rotations model on several quantum processors. For each configuration, the experiment was executed both on the actual machine and classically via simulation of the machine’s noise model. This was done to understand the effect of the QPU’s quantum volume and of its topology on the output, as well as to validate the noise models through simulation. As expected, the circuit sizes (especially in terms of depth) make the effects of decoherence on the results evident for all configurations, which hinders the proper extrapolation of the target measurement. Nevertheless, these findings allow for a baseline for future works focused on providing solutions in this regard, by reducing the circuit depth or exploiting more stable qubits.

The configurations considered involved 2 to 4 assets and 1 to 3 systemic risk factors. The required number of qubits varied from a minimum of 7 to a maximum of 13. The quantum processors used were as follows:**Ibm_perth** and **ibm_lagos**, each with 7 qubits and a quantum volume of 32.**Ibm_canberra** and **ibm_algiers**, each with 27 qubits and quantum volumes of 32 and 128, respectively.

The topology for these architectures is shown in Appendix A.

The aggregate results of the simulated experiments are displayed in Figure 6, while those related to real hardware executions are shown in Figure 7. As mentioned earlier, the depth of the circuit does not allow for the extrapolation of the correct expected value from the computation. For this reason, we deemed it essential to study the effect of noise on the circuit and observe its behavior. To investigate this aspect, we plotted the ratio between the estimated expected loss and the maximum possible loss (which coincides with the sum of the LGDs of the various counterparties) on the x-axis. It should be noted that we used the expected loss as the output metric, estimated directly using the objective qubit.

The complete and non-aggregate results, as well as the code used to generate them, are available in a public repository [28].

## 4. Discussion

Noise is one of the major challenges facing quantum computing, as it can cause errors in qubits. The sources of noise can vary, from environmental factors such as temperature and electromagnetic radiation to decoherence and imperfections in the hardware itself. As a result, researchers have been actively investigating ways to characterize and mitigate the impact of noise on qubits [29,30,31,32].

What we observed, both by simulating quantum machines with their respective noise and by directly performing experiments on QPUs, was that the estimation of the expected total loss tended to converge towards half of the maximum possible loss (see Figure 6, Figure 7 and Figure 8) regardless of the actual expected result. This configuration would correspond to a scenario in which the default probability of counterparties is exactly 50%. This is related to the loss of information due to the execution exceeding the qubits’ coherence time.

A potentially interesting aspect that emerges from the analysis concerns **ibm_algiers**: this machine with a higher quantum volume (128) shows a greater variance around the central value, a potential indication of how the continuous improvement of this particular dimension suggests a future successful application of this and other algorithms on quantum machines.

### 4.1. Scalability and Complexity

While the proposed multiple-rotations variant of the quantum model presents an advantage over the original implementation [4] in terms of qubits required for small values of *K* and *R*, this advantage disappears when the algorithm scales to a realistic setting with thousands of assets and tens of factors. At this scaling, the overhead derived from the presence of the sum register becomes negligible, as the number of qubits it requires scales logarithmically as $O(\log_{2}\left(\sum_{i=1}^{K}LGD_{i}\right))$. However, using the Qiskit *LinearAmplitudeFunction* register [26] requires one additional qubit for each asset taken into account, thereby doubling the increase in terms of qubits that each additional asset entails. From a practical perspective, this translates into an increase in the width of the circuit with respect to the number of assets *K* that approaches O(2K) instead of O(K), which was the rate for the implementation in [4]. Moreover, for both proposed variants, the number of required qubits increases linearly with the number of factors, proportional to $n_{Z}$.

Regarding algorithm complexity, the iterative QAE introduced in [19] and used as a subroutine for our algorithm has a number of queries bounded by
(9)$$\frac{1.4}{\epsilon }\ln\left(\frac{2}{\gamma }\log_{2}\left(\frac{\pi }{4\epsilon }\right)\right),$$
where 1−γ∈(0,1) is the required confidence level and ϵ>0 is the target estimation error. Thus, if we set as an example 1−γ=99.9%andϵ=0.05%, we need around 28 thousand applications of the Grover operator.

The main advantage of IQAE is that it does not increase the number of required qubits, as it does not require performing quantum phase estimation and still provides convergence proofs (which are instead missing for many of the other variants of the original QAE algorithm, such as the one in [33]).

For what concerns the uncertainty model, extrapolating from [4], the standard implementation of U would require first *K* uncontrolled Y-rotations followed by $n_{Z}K*R$ controlled Y-rotations. As in the original analysis, we ignore the preparation of $U_{Z}$ as it can be performed efficiently and does not depend on K. It is important to underline the possibility of implementing U more efficiently by duplicating the Z-qubits *w* times. Multiple copies of *Z* allow us to parallelize the preparation of the qubits representing the counterparties, achieving a depth of (nZK∗R)/w controlled Y-rotations. For further information and analysis, we refer the reader to [12] and particularly to [4], which contains an exhaustive analysis dedicated to the number of gates required for the original implementation.

For our implementations, we highlight the increase in terms of gates needed due to the use of the *LinearAmplitudeFunction* class. This circuit uses controlled linear rotations and comparator registers to implement the piecewise linear function on qubit amplitudes [26]. The number of such registers (and thus of the required gates) increases as $O(2^{K})$. Thus, we observe a significant increase in terms of circuit depth in order to allow arbitrary values for the LGD parameters. However, alternative methods are already being proposed that can decrease the circuit depth needed for encoding the uncertainty model. In particular, in [34], the authors propose an alternative loading method based on quantum generative adversarial networks with encouraging results in terms of saved quantum resources. Moreover, in [35], a novel promising approximate quantum compiling approach is presented. This method would significantly lower the number of physical operations needed to implement complex quantum operators, such as the *LinearAmplitudeFunction*.

## 5. Conclusions

In this paper, we offered solutions to address the limitations of the quantum credit risk analysis algorithm, making it a more effective tool for future advancements in quantum computing technology. We illustrated our proposal and presented the results of several tests (both on quantum and classical hardware) that show the capabilities of our approach and the remaining challenges in terms of scalability and execution on actual QPUs.

The analysis highlights the need for further improvements in qubit coherence since our proposed measures require significantly more gates and qubits at scale than the previous implementation.

Thanks to the improvements proposed, our architecture can take non-integer values for the LGD vector, increasing input flexibility and allowing the use of real-world data. Additionally, our new uncertainty model with multiple risk factors corresponds to the framework most commonly used by big entities in the financial sector [36,37]. These enhancements allow for the creation of new benchmarks for the quantum model. These benchmarks should aim to enable a fair comparison with the classical algorithms currently used by financial institutions and, most importantly, will be able to use the same data for accurate comparison.

In conclusion, it is important to mention that while the proposed quantum credit risk model has the potential to improve the classical CRA process, its integration into production environments will require further research and development. In particular, the integration with existing data pipelines and possibly the design of an end-to-end digital twin will be necessary to evaluate the performance of the quantum model. Additionally, new regulations and legal requirements may be needed for the adoption of quantum algorithms in sensitive financial applications. These considerations highlight the importance of continued collaboration between researchers, financial institutions, and regulatory bodies to ensure the responsible and effective deployment of quantum technologies in the financial industry.

## Figures and Tables

**Figure 1 entropy-25-00593-f001:**
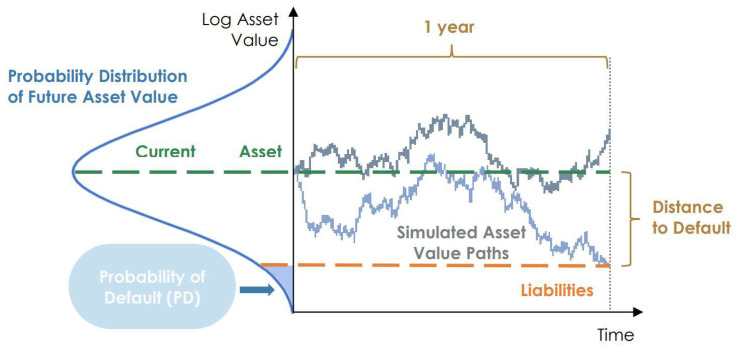
Asset-based default model.

**Figure 2 entropy-25-00593-f002:**
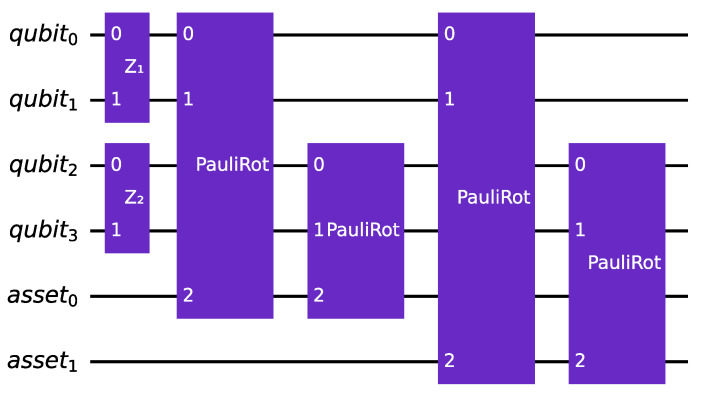
An instance of the multi-factor version of the quantum circuit that encodes the canonical uncertainty model, using multiple rotations. The example involves K=2 assets and $n_{z}=2$, which means that two qubits are used to encode each normal standard distribution. The example also takes into account two risk factors (R=2).

**Figure 3 entropy-25-00593-f003:**
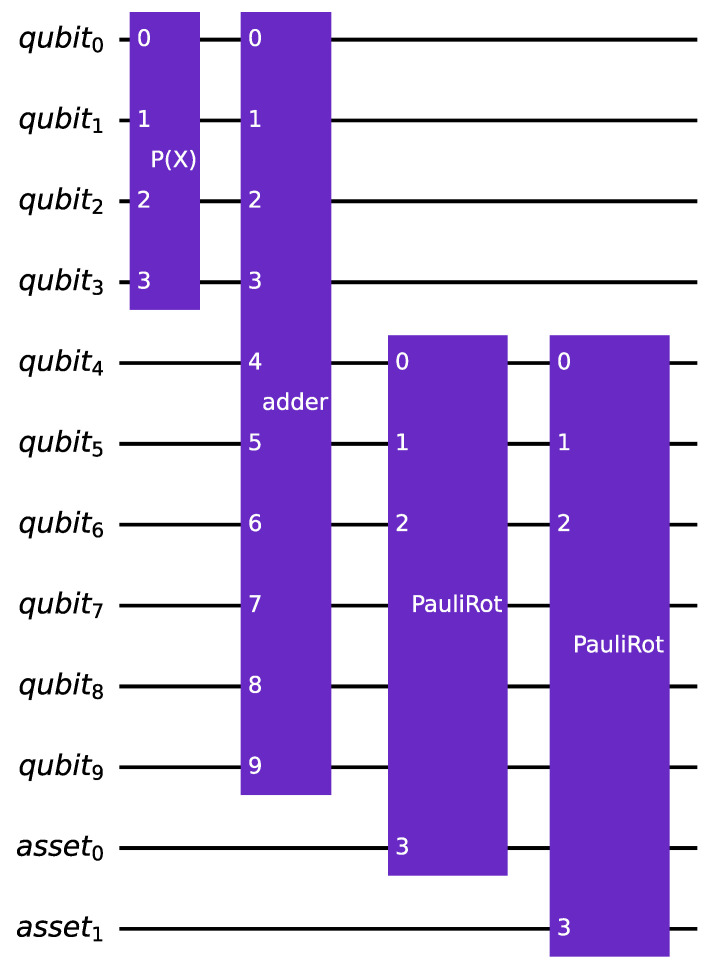
An instance of the multi-factor version of the quantum circuit that encodes the canonical uncertainty model. It has identical parameters to the circuit illustrated in Figure 2 but uses only one rotation per asset.

**Figure 4 entropy-25-00593-f004:**
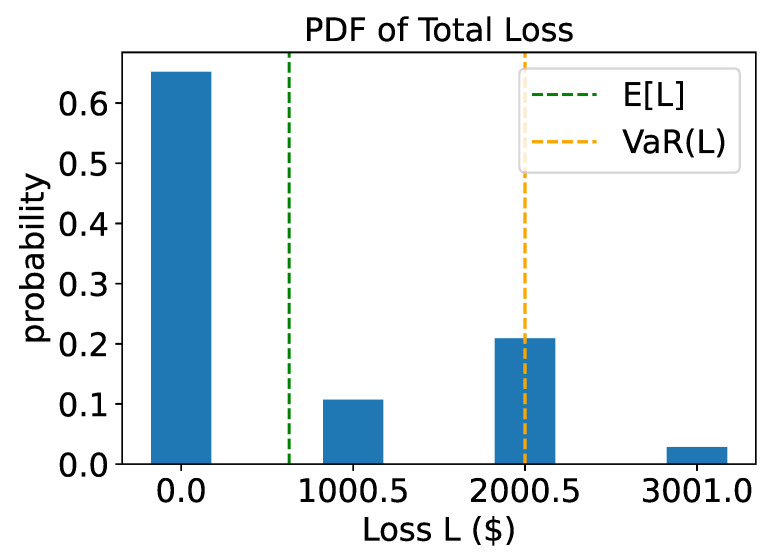
Noiseless simulation: probability distribution function of total loss. The green dashed line shows the expected loss while the orange dashed line shows the value at risk.

**Figure 5 entropy-25-00593-f005:**
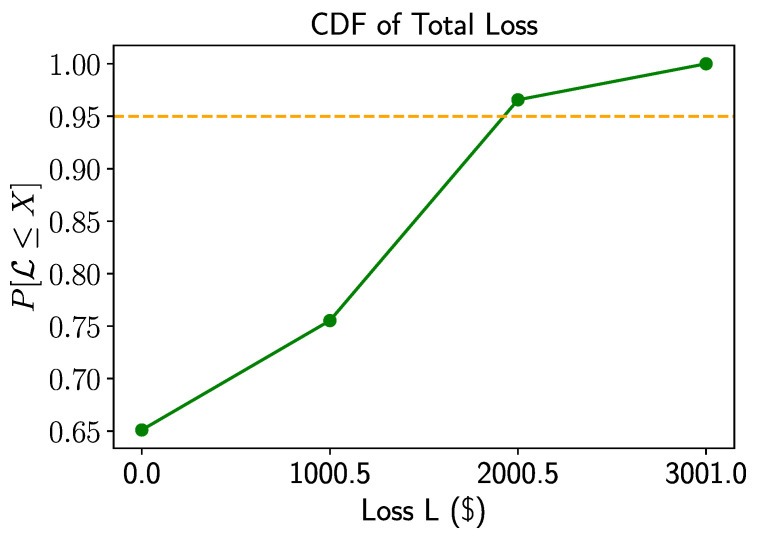
Noiseless simulation: CDF of total loss L in green and target level of 95 percent in orange.

**Figure 6 entropy-25-00593-f006:**
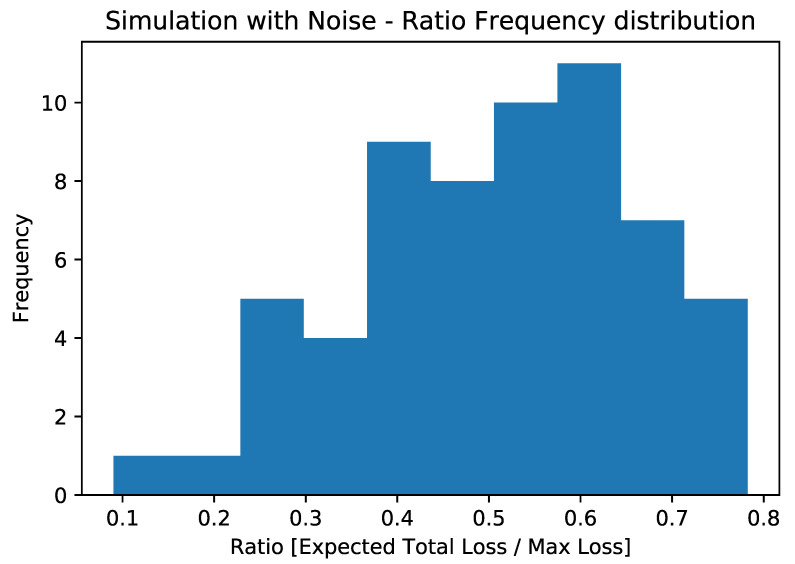
Ratio frequency distribution for the experiments conducted on classical machines, simulating the effects of noise thanks to noise models from the quantum experiments.

**Figure 7 entropy-25-00593-f007:**
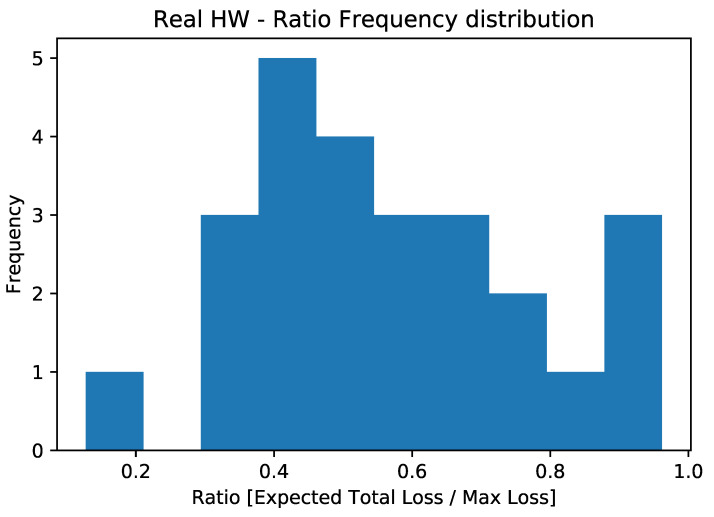
Ratio frequency distribution for the experiments conducted on quantum hardware.

**Figure 8 entropy-25-00593-f008:**
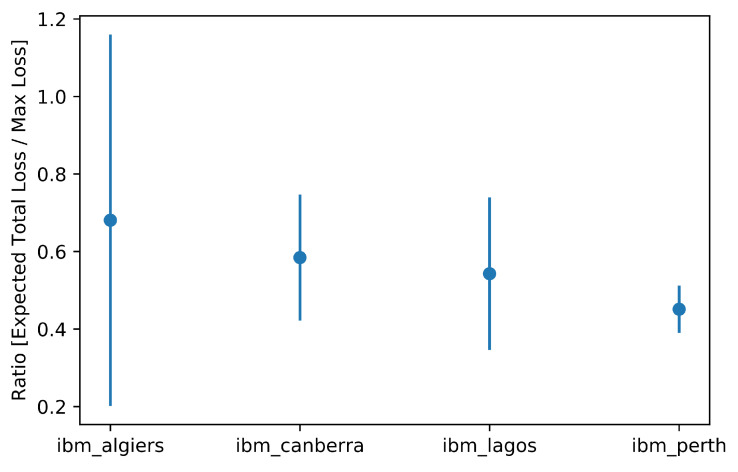
Detailed representation of results on real hardware by architecture.

**Table 1 entropy-25-00593-t001:** Problem parameters for the two-asset example (noiseless simulation).

Asset Number	Loss Given Default	Default Prob.	Sensitivity	Risk Factor Weights
k	${\mathrm{LGD}}_{k}$	$p_{k}^{0}$	$\rho_{k}$	${\left(\alpha_{1},\alpha_{2}\right)}_{k}$
1	1000.5	0.15	0.1	0.35,0.2
2	2000.5	0.25	0.05	0.1,0.25

## Data Availability

The data presented in this study are openly available in QVaR at https://doi.org/10.5281/zenodo.7729267, reference number [28].

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
