# Peer review of "A More General Quantum Credit Risk Analysis Framework"

_entropy, 2023, doi:10.3390/e25040593_

Round 1

Reviewer 1 Report

The work is devoted to the actual topic of the development of quantum mathematics and quantum algorithms in relation to the problems of credit risk management.

The paper is written in a competent scientific language, the references fully reflects the current research in the field.

The article can certainly be recommended for publication in the journal, with some adjustments:

1. Figures and most of the text in chapter 2.1, 2.2 repeat the figures and text of cited article [1]. It is recommended to revise these sections.

2. In the section of discussion of results, it is desirable to show exactly how quantum computing and the developed algorithms can be built into the financial scoring system of banks. How is it aligned with the Basel II standards. How would the proposed algorithms be integrated into the existing data analysis systems of banks that do not use quantum technologies ?

3. How is it possible to design a digital twin of a banking risk management system and integrate quantum algorithms into it?

Reviewer 2 Report

In this manuscript, the authors propose a more general quantum credit risk analysis framework, which is an extended version of their previous work. As we known, near-term quantum computing devices (such as superconducting quantum processor [Science China Information Sciences 63, 180501 (2020)]) have the capability to enable some applications, as mentioned in the recent review paper [arXiv:2211.08737 (2022)]. Credit risk analysis is an interesting application for quantum computing. I could recommend the publication of this manuscript in Entropy after the authors address the following comment.

The proposed algorithm is claimed to have quadratic speedup over classical analogous methods. Could the authors provide a detail complexity and error analysis of their algorithm?
